# The Intertwining of Posttraumatic Stress Symptoms, Alcohol, Tobacco or Nicotine Use, and the COVID-19 Pandemic: A Systematic Review

**DOI:** 10.3390/ijerph192114546

**Published:** 2022-11-05

**Authors:** Amaury C. Mengin, Julie Rolling, Christelle Porche, Amaury Durpoix, Laurence Lalanne

**Affiliations:** 1Psychiatry, Mental Health and Addictology Department, Strasbourg University Hospital, 67000 Strasbourg, France; 2Regional Center for Psychotrauma Great East, 67000 Strasbourg, France; 3Institut National de la Santé et de la Recherche Médiale, Unité de Recherche 1114 (INSERM U1114), Cognitive Neuropsychology, and Pathophysiology of Schizophrenia, 67000 Strasbourg, France; 4Centre National de la Recherche Scientifique Unité Propre de Recherche 3212 (CNRS UPR 3212), Institute for Cellular and Integrative Neurosciences (INCI), 67000 Strasbourg, France; 5Faculty of Medicine, Strasbourg University, 67000 Strasbourg, France; 6Fédération de Médecine Translationnelle de Strasbourg, 67000 Strasbourg, France

**Keywords:** posttraumatic stress disorder, posttraumatic stress symptoms, alcohol, nicotine, tobacco, COVID-19, pandemic

## Abstract

Background: Posttraumatic stress symptoms (PTSSs) and alcohol, tobacco, or nicotine use are frequently associated conditions. The COVID-19 pandemic has been a stressful situation globally and has worsened mental health conditions and addictions in the population. Our systematic review explores the links between PTSSs and (1) alcohol use and (2) tobacco or nicotine use during the COVID-19 pandemic. Methods: We searched the PubMed, PsycINFO, and Web of Science databases for studies published between January 2020 and 16 December 2021. We included studies published in English concerning adults or adolescents. Included articles dealt simultaneously with the COVID-19 pandemic, PTSSs, and alcohol, tobacco, or nicotine use. The reports included were cross-sectional, longitudinal, or cohort studies. We categorized the reports according to the population explored. Our main outcomes are the impacts of PTSSs on (1) alcohol use and (2) tobacco and nicotine use and their relation to COVID-19-related stressors (worries, exposure, lockdown, and infection, either of self or relatives). Results: Of the 503 reports identified, 44 were assessed for eligibility, and 16 were included in our review, encompassing 34,408 participants. The populations explored were the general population, healthcare workers, war veterans, patients with substance use disorders, and other vulnerable populations. Most studies were online surveys (14) with cross-sectional designs (11). Every study explored alcohol use, while only two assessed tobacco use. In most populations explored, a high level of PTSSs was associated with alcohol use increase. COVID-19-related stress was frequently correlated with either high PTSSs or alcohol use. In healthcare workers, PTSSs and alcohol use were not associated, while COVID-19 worries were related to both PTSSs and alcohol use. Discussion: 1. PTSSs and increased alcohol use are frequently associated, while COVID-19 worries might trigger both conditions and worsen their association. Alcohol use increase may represent either an inadequate way of coping with PTSSs or a vulnerability amid the COVID-19 pandemic, leading to PTSSs. As most studies were cross-sectional online surveys, longitudinal prospective studies are needed to ascertain the direction of the associations between these conditions. These studies need to be sufficiently powered and control for potential bias and confounders. 2. Our review highlighted that research about PTSSs and tobacco or nicotine use is scarce.

## 1. Introduction

### 1.1. Co-Morbidity between Substance Use Disorder (SUD) and Posttraumatic Stress Disorder (PTSD)

Substance use disorder (SUD), a comorbidity of posttraumatic stress disorder (PTSD) [1,2], is common, including in teens [3]. In the general population, 21.6% to 43.0% of people with PTSD suffer from comorbid SUD, compared to 8.1% to 24.7% of people without PTSD (2). One possible explanation is that substance use may help patients to deal with PTSD symptoms (PTSSs) (e.g., flashbacks, nightmares, avoidance, negative beliefs and emotional state, hyperarousal, irritability, and sleep disturbances). This coping strategy, especially alcohol and opioid consumption, also exists following highly stressful events, such as childhood adverse events or repetitive traumatic events [4,5]. In parallel, PTSD is more frequent in people with SUD, up to 42.5%. On the one hand, this population is overexposed to traumatic events (e.g., car or drug-related accidents, exposure to violence) [2]. On the other hand, SUD is associated with impulsive reactions and misinterpretations, leading to maladaptive behaviors and fights [6,7,8]. Finally, both disorders may share common factors, whether genetic, neurobiological, or environmental [9]. The most common substances misused are tobacco (by 24% of patients with PTSD [10]) and alcohol (by 9.8% to 61% of patients with PTSD [11]).

### 1.2. COVID-19 Pandemic as a Stressful or Traumatic Event, and COVID-19, Alcohol, Tobacco, and Nicotine Consumption

The characterization of the pandemic as a stressful or traumatic event is still under discussion [12]. Nevertheless, the COVID-19 pandemic has been a long-lasting and unpredictable stressful event. Nearly one in four adults has experienced significant stress levels during the pandemic [13]. This global event comprises both the presence of the virus and the measures taken to prevent it, including restrictions such as lockdowns, which have led to increased loneliness, sedentary lifestyles, and disrupted habits (sleep, food and drug consumption, physical activity) [14,15,16]. The level of exposure varies according to the individual (e.g., healthcare workers or relatives of resuscitated patients) and to the time of exposure. Thus, COVID-19 may be characterized as a traumatic event for a part of the population and as a stressful event for others. Among the factors associated with PTSSs in the general population, frequent exposure to news about COVID-19, COVID-19-specific worries, quarantine, and one’s own COVID-19 infection are frequently reported [17,18,19]. A high level of exposure to COVID-19 among healthcare workers in ICU or emergency units has also been associated with higher levels of PTSSs [20]. In 2020, concerns were raised about increased alcohol consumption since the beginning of the pandemic [21]. Studies later confirmed the impacts of lockdowns on alcohol consumption in the general population and on patients with a pre-existing alcohol use disorder, causing either increased alcohol intake or relapse [6,22,23]. COVID-19-related PTSSs might play a mediating role in the increased alcohol consumption. In parallel, alcohol consumption might be an avoidance strategy or foster isolation, self-deprecation, and guilt (i.e., PTSSs). Fewer studies have explored tobacco or nicotine consumption, revealing an increase during the lockdown associated with ages 18–34, high educational levels, and anxiety [24,25]. Table 1 summarizes these findings in the existing literature.

### 1.3. Interactions between PTSD Symptoms (PTSSs), Alcohol, Tobacco, or Nicotine Use and COVID-19-Related Stress Factors

As stated above, to our knowledge there is currently no literature review exploring the links between PTSD symptoms and alcohol, tobacco, or nicotine consumption during the COVID-19 pandemic. This review aims to evaluate the impacts of the COVID-19 pandemic on PTSD symptoms and on alcohol, tobacco, or nicotine consumption and to explore the specific interactions between these conditions since the beginning of the pandemic, in both the general population and in specific vulnerable populations: people with pre-existing trauma (war veterans), people with pre-existing SUD, most-exposed healthcare workers, and other vulnerable populations.

Our main outcomes are the impact of PTSSs on (1) alcohol use and (2) tobacco and nicotine use in relation to COVID-19-related stress factors (worries, exposure, lockdown, and self- or relatives’ infection).

We will also summarize the impact of these conditions on public mental health during the COVID-19 pandemic.

## 2. Materials & Methods

### 2.1. Global Methodology

#### Protocol and Registration

This systematic review was executed according to PRISMA guidelines for systematic reviews (see Appendix A) [27,28]. This systematic review was not preregistered.

### 2.2. Information Sources

One of the authors (C.P.) screened three databases: PubMed, PsycINFO, and Web of science. PubMed was last consulted on 14 December 2021, while PsycINFO and Web of science were last searched on 16 December 2021. We also consulted Cairn, but this database was not found to be relevant to this study’s topic. The keywords were defined according to the most frequent keywords used in studies of PTSD and COVID-19. Regarding addiction medicine, the authors chose words corresponding to substances specific to the scope of the present study but did not screen every SUD.

We used a combination of keywords relevant to the three categories studied: COVID-19, trauma, and alcohol, tobacco, or nicotine addiction. For COVID-19, we used the terms “coronavirus,” “COVID-19,” and “SARS-CoV-2.” For trauma, we used the keywords “PTSD,” “trauma,” and “psychotrauma.” Finally, for addictions, we used “cigarette,” “tobacco,” “nicotine,” and “alcohol.” Then, we conducted 36 (3 × 4 × 3) searches in each database. We did not use any date restrictions. The detailed results of our database search are available in Appendix A. One researcher (C.P.) reviewed the titles and abstracts of the records found in the databases; we did not use automated search tools. Then, two researchers (A.M. and A.D.) independently screened the titles and abstracts of all articles retrieved. A consensus was reached between all authors concerning which articles to screen full text. A.M. and A.D. then screened full texts of these articles. C.P. and J.R. were consulted in cases of disagreement.

Eligible variables were categorized as follows:→COVID-19-related stress measurement.→PTSS or PTSD measurement.→Alcohol, tobacco, or nicotine consumption measurement.
○Consumption level at the time of assessment.○Changes in this consumption over time.○Existence or non-existence of SUD.

We examined the links between these variables.

We also collected data on the following:→The report: authors, country, date of publication.→The study: time of participant recruitment, study design.→The population: mean age, gender, general and specific populations, level of exposure to COVID-19, and history of psychiatric illness.

### 2.3. Eligibility Criteria

The criteria used for including a study in our review were as follows. We included all investigations of adults and adolescents (≥12 years old), whether they involved the general population or specific populations (e.g., healthcare workers and veterans). The participants must have been recruited during the COVID-19 pandemic (i.e., from January 2020 in China and March 2020 in other countries). To be included in our review, articles had to simultaneously address COVID-19 and confinement, trauma, and alcohol, tobacco, or nicotine (cigarette or electronic cigarette) consumption. The study designs included were cross-sectional, longitudinal, and cohort studies. We also included interventions assessing means of reducing psychological distress during COVID-19 when they explored our dimensions of interest. We included studies published in English in international journals, regardless of the languages initially used for questionnaires. We excluded studies that did not assess PTSSs or alcohol, tobacco, or nicotine use and those that did not explore the relationships between these variables.

### 2.4. Study Selection

We identified 503 records in our database research. We removed 418 records before the screening. Of the 85 remaining records, 44 were assessed for eligibility.

We excluded 8 records that were not original studies; 2 studies that did not explore PTSD or PTSSs; 17 studies that did not examine the links between alcohol, tobacco or nicotine consumption and PTSD or PTSSs; and 1 study conducted before the pandemic. In all, we included 16 articles in our review, for a total of 34,408 participants (N_min_ = 68, N_max_ = 11,325, *sd* = 3216; see details in Figure 1).

## 3. Results

Among these 17 articles, we identified different populations: the general population, healthcare workers, war veterans, patients with SUDs, and specific vulnerable populations (students, sexual minorities, and low and middle-income adults). All these articles explored alcohol use, while only two addressed tobacco or nicotine use.

### 3.1. General Population

Five articles explored the links between alcohol consumption and PTSSs during the pandemic in the general population [29,30,31,32,33]. Most studies were cross-sectional (4 out of 5; Table 2).

The literature shows a bidirectional relationship between alcohol use and PTSSs. On one hand, Currie et al. explored the links between PTSSs and substance use during Wave 1 (April 2020) in 933 adults in Canada, adapting the five-item Primary Care for PTSD Screen to assess pandemic-related PTSSs [29]. In total, 47.8% of adults reported at least one PTSD symptom, and 15.4% met the criteria for high pandemic-related PTSD symptomatology (3 or more symptoms). Pandemic-related PTSSs were significantly more frequent in younger adults, in those socioeconomically disadvantaged due to low income, and in women. The participants were asked whether their alcohol or cannabis consumption had increased very much, decreased very much, or had not changed during the last month. Significant alcohol use increases were reported by 12.8% of women and 10.7% of men. High levels of pandemic-related PTSSs were associated with a two-fold higher risk of substance use increase of alcohol or cannabis (OR = 2.58, 95% CI [1.43, 4.63] and OR = 2.73, 95% CI [1.41, 5.30]) among women and men, respectively. PTSSs were more frequent in adults with increased substance use. For example, 33.3% of women who had increased their substance use reported nightmares or intrusive thoughts about COVID-19, in comparison to 18% among women who did not (no *p*-value provided by the authors).

Another study shows complementary results. LaRosa et al. explored the links between pandemic-related PTSSs, dissociation, and addictive behaviors among 219 non-clinical Italian adults (69.4% females, mean age = 37.6) [30]. Participants completed the Impact of Event Scale–Revised Version (IES-R) and the Addictive Behavior Questionnaire (ABQ), which assessed different substances and activities. They were then classified as “alcohol-addicted” (58.4%) or “not alcohol-addicted” (41.6%). No statistically significant score differences in PTSSs or dissociation were highlighted in the alcohol-addicted group in comparison to the non-alcohol-addicted group. Interestingly, IES–R scores were not significantly higher in the alcohol-addicted group (*p* = 0.40).

On the other hand, Rousset et al. explored the factors associated with PTSD in 6687 Italian adults during Wave 1 [32]. They used the four-item Startle, Physiological Arousal, Anger, Numbness questionnaire, which was not adapted for COVID-19, and found that 43.8% of participants reported PTSSs. The participants with increased alcohol consumption during the lockdown had a significantly higher risk of presenting PTSD symptoms (57% vs. 44.4%, Adjusted Prevalence Ratio (Adj PR) = 1.19 [1.05–1.34]). Though the authors provided a questionnaire assessing tobacco smoking habits, they did not explore this data further.

In the same way, Makhashvili et al. explored the influence of COVID-19 concerns on mental health disorders and alcohol consumption among 2088 Georgian adults [31]. They measured trauma symptoms with eight items from the International Trauma Questionnaire (ITQ) and assessed drinking as a coping method by asking, “Do you drink alcohol to address your concerns about COVID-19?”. A multivariate regression analysis revealed that a greater level of COVID-19 concern was associated with a higher risk of PTSD (Coefficient = 2.78, 95% CI [1.40, 4.16], *p* < 0.01). Drinking alcohol was associated with a higher risk of PTSD (OR = 1.81, *p* = 0.01). In contrast, effective coping strategies (such as reading, TV, radio, doing housework or do-it-yourself (DIY), exercise, meditation or relaxation exercises, and positive thinking) were negatively associated with PTSD (all *p*s < 0.02).

These results were confirmed by a longitudinal study conducted by Veldhuis et al. using a sample of 1567 US adults between April (T1) and September 2020 (T2). The authors adapted the 15-item IES for COVID-19 to assess PTSSs. Participants were asked if they currently “use alcohol or other drugs to help [them] get through it.” Response options ranged from “not at all” to “a lot.” At baseline, almost 50% of participants reported probable PTSD. Among the 35% of participants who reported using alcohol in the past week, 35.6% reported drinking more than usual. Acute stress symptoms significantly decreased between baseline (Mean (M) = 9.7, Standard Error (SE) = 0.511) and the two-week follow-up (M = 8.5, SE = 0.526, *p* < 0.01). Concerning the factors predictive of PTSD at five months follow-up, participants with baseline PTSD had an eight-times higher risk of presenting with PTSD at five months (adjusted Odds Ratio (aOR) = 7.75, 95% CI [5.87, 10.22]). Using alcohol or other substances to cope was associated with higher odds of PTSD at five months (aOR = 1.21, 95%CI [1.01, 1.46]) compared to participants who did not use substances to manage. However, this relationship was no longer significant when social support was added to the model.

To summarize, four of five studies indicate an association between alcohol consumption and PTSSs during the pandemic. We will now focus on populations vulnerable to PTSD and AUD, specifically, war veterans.

### 3.2. War Veterans

War veterans were already at particular risk for pre-morbid PTSD before the COVID-19 pandemic, due to their previous trauma loads. Exposure to war scenes is also associated with SUD, particularly alcohol use disorder. Therefore, PTSD and AUD might also co-exist in war veterans, with alcohol consumption being a way to cope with flashbacks and other PTSSs. Exposure to a new stressful situation, such as the COVID-19 pandemic, might reactivate or complicate previous PTSD and substance use in such a vulnerable population.

Four studies explored the links between PTSD and alcohol consumption in war veterans during the COVID-19 pandemic [26,34,35,36] (Table 3).

Fein-Schaffer et al. (2021) explored the impact of pre-existing PTSD and AUD on biopsychosocial responses to the pandemic in 101 U.S. military veterans in a longitudinal study [26]. Participants (83.2% male, mean age = 59.3 years) completed clinician-led assessments pre-pandemic and phone-based assessments between May and September, 2020. The response to the COVID-19 pandemic was evaluated by a newly created questionnaire, the Rapid Assessment of COVID-19 Related Experiences (RACE). This instrument consists of 26 items, including the assessment of PTSSs, substance use (alcohol, non-alcohol, or prescribed drugs), COVID-19 exposure, and concern about contracting COVID-19. Pre-pandemic PTSD severity was significantly associated with pandemic-related increases in PTSSs (r = 0.38, *p* < 0.01) but not with substance use. Pre-pandemic AUD severity was significantly associated with both pandemic-related increases in PTSSs (r = 32, *p* = 0.01) and substance use (r = 0.47, *p* < 0.01). During the pandemic, substance use and PTSSs were significantly correlated (r = 0.273, *p* < 0.01), and both were significantly associated with COVID-19 self-exposure (r = 0.227, *p* < 0.05 and r = 0.311, *p* < 0.01, respectively). Only PTSSs were significantly associated with proximal (close) relatives’ COVID-19 exposure (r = 0.289, *p* < 0.01). Pre-pandemic AUD severity significantly predicted self-exposure to COVID-19 (*β* = 0.264, *p* = 0.02). In total, this study showed that pre-pandemic AUD was a vulnerability factor for PTSSs, substance use, and self-exposure to COVID-19. In contrast, PTSD before the pandemic only predicted PTSSs during the pandemic.

Pedersen et al. studied the mental health of 1230 US veterans and their reactions to the pandemic [35]. First, a longitudinal study measured pre-pandemic PTSD and AUD levels, using PCL-5 and AUDIT, one month before the pandemic and during the following year (August 2020, November 2020, and February 2021). Their study revealed that PTSD symptoms slightly increased in the sixth month after baseline but had decreased at the nine-month and twelve-month follow-ups. Participants with AUD had higher pre-pandemic PTSD symptoms than non-AUD participants but had less-steep increases in symptoms over time (Incidence Rate Ratio (IRR) = 1.49, 95%CI [1.38, 1.59]). No differences in PTSD symptoms were found during the follow-up between veterans with and without AUD.

In a second study, Pedersen et al. evaluated responses to the COVID-19 pandemic between the one-month pre-pandemic assessment (T1) and a six-month follow-up (T2) [36]. COVID-19 reactions were measured with 13 items related to emotional, stress, sleep, and relationship reactions to the pandemic, each rated on a four-point Likert scale. Participants with PTSD at Time 1 reported higher composite scores on responses to COVID-19 measures in the six months of the pandemic. For the whole sample, alcohol consumption decreased between Time 1 and Time 2. The participants who screened positive for PTSD at Time 1 reported significantly higher alcohol consumption at Time 2 (more frequent drinking and binge-drinking days, higher number of average drinks per occasion, and maximum drinks on one occasion) than those who did not (*p* < 0.001). There was a significant interaction between PTSD and reactions to COVID-19 regarding alcohol consumption (*p* = 0.013). Thus, the participants who screened positive for PTSD and reported a high level of negative responses to COVID-19 (emotional reactions, stress, sleep, relationships) during the first three months of the pandemic reported drinking the most frequently and the greatest number of drinks. There was no significant interaction for the past three months of the pandemic.

Davis et al. detailed these results, exploring changes in alcohol use in the same panel of US veterans at six-, nine-, and 12-month follow-ups [34]. Overall alcohol use (IRR = 0.98) and binge drinking frequency (IRR = 0.11) significantly decreased among US veterans during this period. Veterans reported around 11 days out of the past-30 days of alcohol use in pre-pandemic measures (February 2020), compared with 8.8 days at six-month follow-up and about 10 days at nine- and twelve-month follow-ups. They also confirmed the results of Pedersen et al., as a high pre-pandemic PTSD level was associated with higher pre-pandemic alcohol use (IRR = 1.09, 95% CI [1.01, 1.17]) and binge drinking (IRR = 1.24, 95% CI [1.13, 1.36]), and a less-steep decline in alcohol use during the pandemic. War veterans presenting with higher levels of negative reactions to COVID-19 (emotional reactions, stress, sleep), had higher risks of greater alcohol use and more frequent binge drinking at Time 2 (9 months, November 2020; IRR = 0.29, 95% CI [0.23, 0.35] and IRR = 0.39, 95% CI [0.32, 0.46], respectively) and Time 3 (12 months, February 2021; IRR = 0.10, 95% CI [0.04, 0.16], and IRR = 0.39, 95% CI [0.32, 0.46], respectively. In total, in war veterans, both pre-pandemic PTSD and AUD were predictive of PTSD and increased alcohol consumption during the pandemic. In addition, negative psychological reactions to the pandemic worsened alcohol-use increases in participants with PTSD. While war veterans were at risk of both PTSD and AUD, other populations which had already presented with substance use disorder before the pandemic were at risk of worsening their consumption. We will now explore how COVID-19 reactions and PTSSs interacted with substance use in this population.

### 3.3. Individuals with SUD

We found only two studies that assessed the links between alcohol use and PTSD during the COVID-19 pandemic in patients with SUD; both were cross-sectional (Table 4).

Blithikioti et al. (2021) explored risk factors for impaired mental health during the COVID-19 pandemic in 303 individuals with SUD (61.4% men, average age = 49.3) [37]. Participants completed an online questionnaire between June and July 2020. Previous traumatic events were explored via the Childhood Trauma Questionnaire and the Life Events Checklist, the presence of PTSD via the Davidson Trauma Scale, and substance use via the Alcohol, Smoking, and Substance Involvement Screening Test. Lifetime exposure to trauma was reported in 69% of participants, PTSD in 34.4%, and AUD in 46.9%. During the lockdown, most patients did not report a changed frequency of substance consumption (including alcohol). Regarding alcohol, the change was more often toward reducing consumption (18.9%) than toward increasing it (12.5%). Regarding tobacco use, 9.5% of patients reduced their frequency of use during the lockdown, while only 5.4% reported an increase. While the mental health of 50.5% of patients worsened during the lockdown, histories of trauma (OR = 2.02, 95% CI [1.09–3.62], *p* = 0.02) and alcohol use (OR = 1.49, 95% CI [1–2.2], *p* = 0.05) were associated with a higher risk of impaired mental health during confinement.

Yazdi et al. (2020) investigated the links between alcohol consumption and PTSSs in patients with AUD being treated in an Australian addiction department between April and mid-June 2020 (41.7% were hospitalized, and 58.3% were outpatients) [38]. The 127 patients who agreed to participate in the study (66.9% men, mean age = 49.3 years) were classified into three subgroups by comparing their addiction status before and during the pandemic: abstinent (*n* = 37), relapsed (*n* = 41), and consuming (*n* = 49). Their level of alcohol consumption was measured by the Alcohol Use Disorder Identification Test–Consumption part (AUDIT–C) and the presence of PTSD by the Primary Care PTSD screen for DSM5 (PC–PTSD5). There was a significant correlation between AUDIT–C and PC–PTSD5 (*p* = 0.001). In addition, relapsed patients had higher PTSD scores than did abstinent patients (*p* = 0.01). In contrast, there were no differences in PTSD scores between abstinent and consuming patients (*p* = 0.26) or between relapsed and consuming patients (*p* = 0.50).

To summarize, while the literature reports mainly stable or decreased alcohol use, PTSD may be a risk factor for alcohol relapse in the context of the COVID-19 pandemic. Other populations, especially healthcare workers, were vulnerable due to their exposure to COVID-19 during the pandemic. We will now explore our outcomes in this population.

### 3.4. Healthcare Workers and First Responders

Healthcare workers have been particularly exposed to stress during the pandemic. They have faced tremendous work, numerous deaths, and the fear of contaminating themselves and their loved ones [39]. This exposure could constitute a risk factor for various mental health problems, specifically PTSD and addictions.

Only two studies explored the links between PTSD or PTSSs and alcohol, tobacco, or nicotine consumption in healthcare workers and first responders (Table 5).

Among healthcare workers, Greenberg et al. explored the mental health of 709 staff working in intensive care units (ICUs) in the UK between June and July 2020, using a web-based survey [40]. Their sample comprised 291 (41%) doctors, 344 (49%) nurses, and 74 (10%) other healthcare staff. They used a six-item version of the Posttraumatic Stress Disorder Checklist (PCL-6) and the AUDIT–C questionnaire to assess problem drinking. Forty percent of respondents met the threshold for PTSD and 7% for problem drinking. A significantly higher proportion of nurses met the criteria for PTSD (49% vs. 32% and 27% in doctors and other staff, respectively), while no significant difference appeared between occupations concerning problem drinking. Probable PTSD was negatively correlated with well-being (r = −0.601) and positively with depression and anxiety (*r* = 0.730 and *r* = 0.701, respectively), but there was no correlation between any mental health condition (including PTSD) and problem drinking (*r* = −0.013). Problem drinking was negatively correlated with well-being (*r* = −0.135).

Among healthcare workers and rescue personnel, first responders (including firefighters, emergency medical staff [EMS], and law enforcement officers) were also over-exposed to stressful situations during the pandemic. Vujanovic et al. (2020) interviewed 189 first responders, assessing PTSD by the PCL-5 (adapted to COVID-19) and alcohol use severity by the frequency and level of alcohol use (number of drinks on a typical day when drinking). COVID-19-exposed first responders reported significantly greater alcohol use severity than did non-exposed first responders. In this sample of first responders, 6.8% (13 participants) met the threshold for probable PTSD. PTSD symptom severity was not correlated with alcohol use severity. Trauma load (the number of previous traumatic events faced) was associated neither with PTSD symptom severity nor with alcohol use severity. COVID-19 worry was related to PTSD symptom severity (explaining 45% of the variance) and alcohol use severity. COVID-19 exposure was positively associated with alcohol use severity, while years of service was negatively associated with alcohol use severity. In total, alcohol use was associated with COVID-19 worries and exposure in healthcare workers and first responders, while PTSSs were only associated with COVID-19 concerns. Alcohol use and PTSSs were not associated.

To summarize, neither study revealed any relation between PTSD and AUD in the context of COVID-19 in those specific populations. Other specific populations were vulnerable during the COVID-19 pandemic, due to various factors. We will now explore our outcomes in these populations.

### 3.5. Other Vulnerable Populations

#### 3.5.1. Sexual Minorities

With a higher level of stigma and lower social support, sexual minorities (e.g., those who identify as gay, lesbian, bisexual, or queer) may be more vulnerable to pandemic-related stress. Helminen et al. aimed to estimate whether PTSD symptoms and alcohol consumption were higher during high social stress, such as confinement in sexual-minority women exposed to trauma [42] (Table 6). Between April and August 2020, participants were recruited through internet ads posted on different platforms (e.g., social media and LGBTQ-related online listservs). The 68 participants (mean age = 28.74) who agreed to participate were asked to complete the PTSD Checklist for DSM-5 (PCL-5) and the AUDIT-C. The participants with PCL-5 scores ≥ 33 were considered to have probable PTSD, while those with AUDIT-C scores ≥ 3 were supposed to have probable AUD. In total, 35.3% of participants included by Helminen et al. had probable PTSD, which was significantly higher than the 21% (*p* < 0.01) and 21.9% (*p* < 0.05) of participants found in two similar pre-pandemic samples. In contrast, these results were not superior to the 23.7% and 27.3% of participants with PTSD included in two other studies. Concerning alcohol consumption, 47.1% of participants had a probable AUD, which was statistically higher than the 27.1% (*p* < 0.001) and 28.3% (*p* < 0.01) found in two pre-pandemic samples. The proportion of women reporting heavy episodic drinking was 45.6% (*n* = 31), compared to 9.5%, 25.5%, 33.8%, and 39.3% reported in four previous samples. Finally, 17.6% had PTSD–AUD comorbidity, higher than the 8.5% in a pre-pandemic sample of bisexual women (*p* < 0.05).

#### 3.5.2. Low- and Middle-Income Adults

The COVID-19 pandemic has had a substantial economic impact around the world. Individuals with low or middle incomes, who are potentially more stressed from the baseline by financial issues, may have more difficulty managing pandemic-related stress. Tsai et al. sought to investigate the prevalence of mental health illnesses, including PTSD and AUD, and the psychosocial factors linked to them among this population. The authors also explored the effect of testing positive for COVID-19 on mental health [43] (Table 6). A sample of 6607 U.S. adults with incomes below $75,000/year was recruited between May and June 2020 from participants in an online labor marketplace. Participants were classified according to their COVID-19 test status and completed the standardized AUDIT-C and PCL-5 questionnaires. The results of the PCL-5 were considered to show “COVID-19 era stress symptoms” (CS) if participants selected “the experience of COVID-19” as Criterion A. Among patients testing positive for COVID-19 (*N* = 354), 30.4% had a history of PTSD, 32.6% had a history of AUD, 75.5% screened positive for CS, and 76.9% screened positive for AUD. Among patients testing negative for COVID-19 (*N* = 1819), 10.8% had a history of PTSD, 19.4% had a history of AUD, 32.4% screened positive for CS, and 40.3% screened positive for AUD. Among patients not tested for COVID-19 (*N* = 4434), 6.8% had a history of PTSD, 5.5% had a history of AUD, 12.3% screened positive for CS, and 31.0% screened positive for AUD. Pairwise comparisons revealed that participants who tested positive for COVID-19 presented a significantly higher risk for each of these conditions than did those who tested negative. Furthermore, a history of PTSD predicted the risk of COVID-19-era stress symptoms (OR = 1.80, 95% CI [1.47, 2.20]) but did not predict an increased risk of current AUD (OR = 1.04, 95% CI [0.87, 1.24]). Having a history of AUD was predictive of the risk of current AUD (OR = 1.92, 95% CI [1.62, 2.29]) but not of CS (OR = 1.15, 95% CI [0.95, 1.39]).

#### 3.5.3. Students

Students deal with chronic stress related to normal exam success pressure, uncertainty about the future, and financial insecurity. Additional stressors—with a pandemic and a lockdown forcing one to endure courses remotely and alone at home—can exceed some individuals’ emotional regulation abilities and alter their mental health. Xu et al. aimed to assess mental health risk factors among students experiencing extended school closures related to the COVID-19 pandemic [44] (Table 6). An online survey was conducted from 29 June 2020, to 18 July 2020, among 11,254 participants recruited from 30 universities in the Chinese city of Wuhan. Participants were asked about their alcohol and tobacco consumption and completed the PCL-5 questionnaire. During the pandemic, 5.89% and 3.77% of individuals reported alcohol and tobacco consumption, respectively, and 8.46% reported PTSSs. Participants who consumed alcohol during lockdown were significantly more likely to present with PTSSs (16.59% compared to 7.95% [*p* < 0.01]). The students who consumed tobacco during lockdown were significantly more likely to present with PTSD (17.69% versus 8.10% [*p* < 0.01]).

The three studies involving vulnerable populations fail to reveal a common tendency. While there does not seem to be a relation between PTSSs and AUD in low-middle income adults, sexual minorities seem to be at greater risk of AUD-PTSD co-morbidity. In Chinese students, alcohol and tobacco consumption appear to be associated with PTSD during lockdown.

## 4. Discussion

Our review explored the interactions between PTSSs, alcohol consumption and tobacco/nicotine consumption, and reactions to the COVID-19 pandemic, showing strong interrelations between these three variables among various populations [26,29,30,31,32,33,34,35,36,37,38,40,41,42,43,44]. In the general population, as in the specific vulnerable populations examined, PTSSs and alcohol use (including AUD) frequently potentiated each other, though these results are mixed. In the general population, PTSSs and alcohol use were frequently associated. In war veterans, who are at risk of both PTSD and AUD, both pre-pandemic PTSD and AUD are primarily predictive of PTSD and increased alcohol consumption during the pandemic. In people with SUD, the literature reports mainly alcohol use stability or decrease, and PTSD may be a risk factor for alcohol relapse in the context of the COVID-19 pandemic. In vulnerable populations, we did not find any common tendency. While there does not seem to be a relation between PTSSs and AUD in low-middle income adults, sexual minorities seem to be at greater risk of AUD-PTSD co-morbidity. In Chinese students, alcohol and tobacco consumption appear to have been associated with PTSD during the COVID-19 pandemic. In sum, high levels of negative reactions to COVID-19 are often associated with higher PTSSs or alcohol use. For the relationship between tobacco or nicotine use and PTSSs during the COVID-19 pandemic, our review showed a major lack of studies exploring the associations, preventing us from drawing conclusions.

These observations align with the frequent co-occurrence of PTSD and AUD before the pandemic. Previous studies found AUD to be present among 43% of PTSD patients. Patients with PTSD are about six times more likely to develop AUD than are individuals without PTSD [45]. Similarly to previous psychosocial models of the PTSD–AUD association, the main model arising from our review is the latter: a high level of negative reactions to the COVID-19 pandemic, reflecting high levels of PTSSs and alcohol use as an externalizing avoidance symptom. The study of Makhashvili et al. is particularly representative of this pathway, as it showed alcohol use to be the only coping strategy positively associated with PTSD. In contrast, all other coping strategies, and social support, were negatively associated with PTSD. These observations reveal a need to inform the population about emotions and concerns that might be raised by the pandemic (or other stressful situations) and successful or ineffective ways to cope with them [46]. Interventional studies providing emotional regulation or coping strategies (mindfulness, relaxation, etc.) would also help to precise the efficacy of such methods in preventing PTSSs and alcohol use increase during a stressful event.

In parallel, pre-existent AUD might reveal one’s poor coping strategies and susceptibility to develop PTSD. Previous studies have showed that among veterans with AUD, about 63% suffer from comorbid PTSD, while in veterans with PTSD, about 76% are diagnosed with comorbid AUD [45]. This co-occurrence has also been found among veterans during the pandemic. Moreover, chronological associations have been demonstrated in these populations, as pre-pandemic AUD predicted PTSD during the pandemic, and pre-pandemic PTSD predicted AUD during the pandemic, reciprocally. Similar patterns have been found in individuals with SUD, as the worsening of AUD (relapse) is more obviously linked with PTSSs than is alcohol use stabilization (either abstinence or consumption). These observations confirm the susceptibility of individuals with AUD and PTSD to developing poor coping strategies when enduring a new stressful event. These considerations belong to a larger field of reflections concerning frequently co-occurring psychiatric and addictive disorders, in which psychiatric disorders might prompt an individual to cope via substance use while substance use may precipitate mental health problems [47].

Different patterns of association appeared among healthcare workers, as in both studies exploring these individuals, PTSS severity and alcohol use severity were not significantly associated. In parallel, one study revealed that the level of worry concerning COVID-19 was associated with both PTSS severity and alcohol use severity. In a population directly and constantly at risk of COVID-19 exposure, this threat seems to play a more central role than in the general population. At the same time, the use of alcohol to cope with PTSSs might be more infrequent than in the general population. The latter observation might be explained by a better knowledge of efficient coping strategies (due to individuals’ experience with stressful situations) and of alcohol-related medical risks. A Korean study showed that firefighters used problem-focused coping skills—seeking social support and wishful thinking—more frequently than did the general population [48]. Problem-focused coping and seeking social support were negatively associated with PTSD among these firefighters. As positive coping strategies, social and organizational support may protect against late-onset PTSD among healthcare workers [49,50]. Very few studies have explored alcohol use in healthcare workers during the pandemic.

Before the pandemic, a study of 639 healthcare workers found that 43.8% had an AUDIT-C score of 0, while 9.1% were classified as “high-risk drinkers” [51]. During the pandemic, a study of 9138 Spanish healthcare workers found that a total of 6.2% reported substance use, compared to the 7% of Intensive Care Unit (ICU) workers with problem drinking found by Blithikioti et al. [37,52]. These rates tend to be similar over time (e.g., before vs. during the pandemic) and across different populations of healthcare workers. Finally, significant differences appeared in PTSD symptoms between the populations explored by Greenberg et al. (40% of probable PTSD among ICU healthcare workers) and Vujanovic et al. (6.8% of PTSD among first responders). These differences might be the result of exposure to COVID-19 and its tragic reality of causing ICU workers to witness daily numerous deaths.

Tobacco and nicotine were rarely explored in the studies examined in our review, though we thoroughly searched for papers exploring such consumption. The only research we reviewed was conducted among Chinese students and found a significantly higher risk of PTSD among smoking students. Recent studies in representative panels of US adults showed that past traumatic events and PTSD are associated with greater smoking prevalence and more intense smoking [53,54]. During the COVID-19 pandemic, more frequent tobacco use was associated with depression, anxiety, stress, and insomnia [55,56]. These results suggest that an association between nicotine and tobacco use would also be associated with PTSD symptoms in the general population during the pandemic. Researchers should investigate these associations, as tobacco or nicotine use might be markers of poor psychological adaptation to a stressful event.

Along with students, adolescents have been particularly impacted by the COVID-19 pandemic, as illustrated in the French population [57]. This phenomenon can be partly explained by school demands and changes in the organization of courses, but also by uncertainty about the future. We did not find any study assessing the association between PTSD and adolescent substance use during the COVID-19 pandemic. However, a report evaluating suicidal risk showed an association between a higher likelihood of suicidal behavior and depression, anxiety, alcohol and tobacco use, and childhood trauma. More research on PTSD and substance use is needed for adolescent populations with known brain vulnerabilities to substance use [58].

There are general limitations to the studies explored in our review. First, numerous studies were cross-sectional and did not provide data from the pre-pandemic period, except those concerning war veterans. This lack limits our conclusions concerning the temporal relationship between alcohol or nicotine use and PTSD and the mechanisms underlying the intertwining of these conditions. Second, most studies were based on online self-report questionnaires, and face-to-face clinician assessments were rarely used, which possibly creates assessment biases impacting the results. Moreover, some questionnaires were newly created and had not undergone a validation process. This assessment bias is current in the COVID-19 literature, as many surveys were provided online due to social contact restrictions. Third, many studies assessing substance use did not separate the different substances (e.g., alcohol, tobacco, and cannabis), preventing us from determining more precisely which substances were associated with PTSSs. These findings invite researchers to differentiate the substances used in future investigations to study the specific role of alcohol use and, even more, tobacco or nicotine use. The same phenomenon has occurred with PTSD, as some studies have merged psychiatric diagnoses, exploring “mental health” or “internalizing disorders” rather than PTSD, depression, and anxiety disorders. Finally, the heterogeneity of the populations studied in the reviewed articles, the measuring methods used, and even the definitions of PTSD symptoms and alcohol consumption, call for necessary caution regarding the findings we have presented.

In our review process, we chose not to use automatization tools to select relevant articles, which might have led to selection bias, and we doubled the review of the pre-selected reports to correct this bias. We decided to focus on papers strictly assessing the links between our main variables (i.e., alcohol, tobacco or nicotine consumption and PTSD symptoms), which prevented us from including reports that might have raised relevant results. However, this strategy allowed us to focus on our primary outcomes and address more specific research questions.

## 5. Conclusions

To conclude, our review draws important insights into the intertwining of PTSD and alcohol, tobacco, or nicotine consumption in the COVID-19 era among many different populations. While we highlighted a lack of studies in the field of tobacco and nicotine consumption, we showed that alcohol consumption and PTSSs were frequently associated in the general population. The relationship seems to be more complex in vulnerable populations.

As COVID-19 exposure and fear are often related to both PTSSs and alcohol consumption, the information provided to the public concerning the means of both reducing COVID-19 risk and of coping with stressful situations (e.g., choosing physical activity rather than alcohol) needs to remain and to be renewed. People affected by COVID-19 would benefit from specific information and support on how to preserve their mental health. Populations already vulnerable to PTSD and AUD before the pandemic need to be monitored carefully, as these conditions potentiate each other. Proactive mental health care for these populations in stressful situations (e.g., phone calls, teleconsultations, and information on mental health services) is also encouraged. More research is needed concerning the mechanisms underlying the interweaving of PTSD symptoms and alcohol, tobacco, or nicotine consumption (e.g., to understand whether alcohol consumption is a way of coping or a vulnerability factor), especially longitudinal prospective studies. Furthermore, research on the impact of tobacco or nicotine on PTSD symptoms during a stressful global event is needed, as the literature on this topic is scarce. Finally, the role of social support in decreasing the risk of PTSD and AUD was raised by some studies and is a core protective factor that needs to be embraced.

## Figures and Tables

**Figure 1 ijerph-19-14546-f001:**
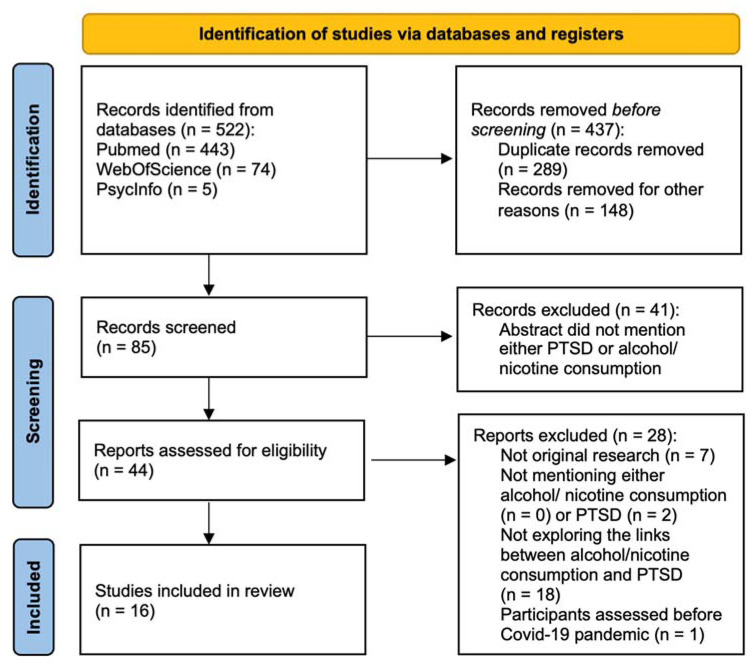
Flow-chart diagram.

**Table 1 ijerph-19-14546-t001:** Previous findings concerning PTSSs, alcohol, tobacco and nicotine consumption, and COVID-related factors.

Outcomes	Previous Findings
Association between COVID-19 related stressors and PTSSs [17,18,19]	Associated with PTSSs: High level of exposure to news about COVID-19 COVID-19-specific worries Quarantine High risk of COVID-19 Own COVID-19 infection High exposure to COVID-19 (healthcare workers)
Association between COVID-19 related stressors and alcohol, tobacco, or nicotine use [6,22,23,24,25]	Lockdown resulting in increasing alcohol and tobacco or nicotine use
Association between PTSSs, alcohol, tobacco, or nicotine use before the pandemic [10,11]	Tobacco use: 24% of patients with PTSD Alcohol misuse: 9.8 to 61% of patients with PTSD
Association between PTSSs and changes in alcohol, tobacco, or nicotine use during the COVID-19 pandemic	To be explored
Association between pre-existing PTSD and COVID-19-related PTSSs [26]	Pre-existing PTSD predicts COVID-19-related PTSSs among war veterans
Association between pre-existing PTSD and changes in alcohol, tobacco, or nicotine use during the COVID-19 pandemic	To be explored

**Table 2 ijerph-19-14546-t002:** Summary of the main data of the studies considered in this review—General population.

Study	Objectives	Design	Period	Population	Country	Measurement Tools	Results
Currie et al. (2021) [29]	Exploring the links between pandemic-related PTSSs and substance use	Cross-sectional derived from a national panel online survey	June 2020	933 community-based adults without pre-pandemic PTSD diagnosis	Canada	Primary care PTSD screen; alcohol/cannabis use increase or decrease during the last month	15.4% met criteria for PTSD symptomatology. High level of pandemic-related PTSSs is associated with a >2-fold increase in the odds of an increased substance use.
La Rosa et al. (2021) [30]	Exploring the prevalence of post-traumatic distress, alexithymia, dissociation, and addictive behavior	Cross-sectional online survey	Between 22 March and 4 May 2020	219 non-clinical adults	Italy	IES-R, DES-II (dissociation), ABQ (addictive behavior)	Mean IES-R score: 30.56 ± 1.40. 58.4% of alcohol-addicted during lockdown. Alcohol addiction is neither associated with PTSSs nor with dissociation.
Makhashvili (2020) [31]	Exploring the influence of concern about COVID-19 on mental health conditions	Cross-sectional internet-based study	Between 25 May and 25 June 2020	2088 adults	Georgia	ITQ, alcohol use evolution (decrease, stable, increase), PHQ-9, GAD-7	11.8 and 12.5% of PTSD among women & men respectively. PTSD is associated with concern about COVID-19. Drinking alcohol to cope is associated with PTSD.
Rousset et al. (2021) [32]	Evaluating the factors associated with PTSD	Cross-sectional self-administered online questionnaire	Between 21 April and 7 June 2020	6687 adults	Italy	SPAN (PTSD symptoms), increased- decreased- unchanged alcohol use	43.8% of participants report PTSSs. Prevalence of PTSSs decreases with age. PTSSs are higher in adults with increased alcohol consumption.
Veldhuis et al. (2021) [33]	Assessing risk factors associated with psychological factors and their changes over time	Longitudinal online survey	From April to September 2020	1567 adults	USA	IES adapted to COVID-19, use of substance to cope on a Likert scale, COVID-19 worries on a 0 to 100 scale, GAD-7	PTSD at baseline and using substances to cope are associated with a higher risk of PTSD at 5-month follow-up.

Abbreviations: ABQ Addictive Behavior Questionnaire, DES-II Dissociative Experience Scale 2nd version, GAD-7 Generalized Anxiety Disorder-7, IES-R Impact of Event Scale-Revised, ITQ International Trauma Questionnaire, PCL-5 PTSD Checklist for DSM-5, PHQ Patient Health Questionnaire, and Startle, Physiological Arousal, Anger, Numbness (SPAN) questionnaire.

**Table 3 ijerph-19-14546-t003:** Summary of the main data of the studies considered in this review—War veterans.

Study	Objectives	Design	Period	Population	Country	Measurement Tools	Results
Davis et al. (2021) [34]	Exploring the changes in alcohol use and binge drinking in relation to pre-pandemic PTSD and AUD.	Longitudinal online questionnaires.	T1 = February 2020, T2 = August 2020, T3 = November 2020, T4 = February 2021	1230 U.S. Veterans	USA	PCL-5, days drinking or binge drinking in the last 30 days	Substantial decrease in alchol use & binge drinking during the pandemic. Higher overall rates of use and smaller decreases of use in those with pre-existing PTSD.
Fein-Schaffer et al. (2021) [26]	Assessing the impact of pre-existing PTSD and AUD on biopsychosocial responses to the pandemic.	Longitudinal pre-pandemic clinical assessment, phone interview during the pandemic.	May-September 2020	101 US Veterans	USA	CAPS-5, SCID	Pre-pandemic PTSD and AUD were associated with pandemic-related PTSD. Pre-pandemic AUD was associated with pandemic-related PTSD and self-exposure to COVID-19.
Pedersen et al. (2021) [35]	Examining changes in PTSD, depression, and anxiety symptoms before and through 12 months after the pandemic outbreak. Examine changes in mental health symptoms among veterans with pre-existing AUD or SUD.	Longitudinal online questionnaires.	T1 = February 2020, T2 = August 2020, T3 = November 2020, T4 = February 2021	1025 U.S. Veterans	USA	PCL-5, AUDIT, PHQ-8, GAD-7	22.8% had pre-COVID PTSD, 52.4% had pre-COVID AUD. Veterans with probable pre-COVID AUD had higher pre-COVID PTSD symptoms. No difference in PTSD symptoms during the pandemic between those with and without AUD.
Pedersen et al. (2021) [36]	Investigating the relationship between PTSD and current reactions to COVID-19 on alcohol and cannabis use.	Longitudinal online questionnaires.	T1 = February 2020, T2 = August 2020	1230 U.S. Veterans	USA	PCL-5, days of alcohol use and binge drinking in the past 30-days	Decrease in alcohol use between T1 and T2. At T2, veterans with PTSD reported 28% more drinking days. Veterans reporting both PTSD and negative COVID-19 reactions drank more frequently, and with greater average drinks per occasion.

Abbreviations: AUDIT Alcohol Use Disorder Identification Test, CAPS-5 Clinician-Administered PTSD Scale for DSM-5, GAD-7 Generalized Anxiety Disorder-7, PCL-5 PTSD Checklist for DSM-5, PC-PTSD Primary Care-PTSD Scale, PHQ Patient Health Questionnaire, SCID Structured Clinical Interview for DSM Disorders.

**Table 4 ijerph-19-14546-t004:** Summary of the main data of the studies considered in this review—Individuals with SUD.

Study	Objectives	Design	Period	Population	Country	Measurement Tools	Results
Blithikioti et al. (2021) [37]	Identifying risk factors associated with mental health outcomes.	Cross-sectional online survey.	June–July 2020	303 individuals with SUD consulting in an addiction unit	Spain	CTQ, LEC, ASSIST, DTS	34.4% have PTSD, 46.9% have AUD, 18.9% decreased alcohol consumption, and 12.5% increased. AUD and PTSD increased the clinical deterioration due to COVID-19.
Yazdi et al. (2020) [38]	Investigating addiction, PTSD symptoms, and COVID-19 factors associated in patients with AUDs.	Cross-sectional face-to-face or telephone interview.	April–June 2020	127 inpatients (41.7%) or outpatients with AUD	Austria	PC-PTSD, AUDIT-C	31.7% have PTSD symptoms due to COVID-19, positive correlation between AUDIT-C and PC-PTSD5, significant difference only between abstinent and relapsed patients.

Abbreviations: ASSIST Alcohol, Smoking and Substance Involvement Screening Test, AUDIT-C Alcohol Use Disorder Identification Test-Concise, CTQ Childhood Trauma Questionnaire, DTS Davidson Trauma Scale, LEC Life Event Checklist, PC-PTSD Primary Care-PTSD Scale.

**Table 5 ijerph-19-14546-t005:** Summary of the main data of the studies considered in this review—Healthcare workers.

Study	Objectives	Design	Period	Population	Country	Measurement Tools	Results
Greenberg et al. (2021) [40]	Identifying the rates of probable mental health disorders in ICU workers.	Cross-sectional web-based survey.	June–July 2020	709 ICU workers	England	PCL-5 6-items, AUDIT	40% probable PTSD, 7% problem drinking, no correlation between PTSD and problem drinking.
Vujanovic et al. (2020) [41]	Exploring the impact of COVID-19 pandemic on first responders’ mental health.	Cross-sectional web-based survey.	June–August 2020	189 first responders	USA	PCL-5, Mental Health Correlates Questionnaire (MHCQ)	6.8% probable PTSD, higher risk of probable PTSD among COVID-19-exposed first responders. No correlation between PTSD severity and alcohol use severity.

Abbreviations: AUDIT Alcohol Use Disorder Identification Test, ICU Intensive Care Unit, PCL-5 PTSD Checklist for DSM-5.

**Table 6 ijerph-19-14546-t006:** Summary of the main data of the studies considered in this review—Other vulnerable populations (students, sexual minorities, and low- and middle-income adults).

Study	Objectives	Design	Period	Population	Country	Measurement Tools	Results
Helminen et al. (2021) [42]	Estimating whether PTSD and hazardous drinking were heightened during high social stress.	Cross-sectional online self-report questionnaire.	April–August 2020	68 trauma-exposed sexual minority women	USA (New York)	PCL-5, AUDIT	35% of probable PTSD, 47% of probable AUD. Higher PTSD symptom severity, probable PTSD, and hazardous drinking indicators compared to similar samples.
Tsai et al. (2021) [43]	Comparing the prevalence of mental illness and alcohol use disorder, and identify factors associated (whose testing positive for COVID-19).	Cross-sample investigation Online survey.	May–June 2020	6607 adults with low and middle-income	USA	PCL-5, AUDIT-C, GAD-2, PHQ-2	20.2% of PTSD, AUD elevated if COVID infection (76.9% if COVID +, 40.3% if COVID −, 30.3% if no test).
Xu et al. (2021) [44]	Evaluating the impact of the long-term COVID-19 pandemic on psychological status among university students who experienced isolation in early days of the pandemic.	Cross-sectional online self-report questionnaire.	June–July 2020	11,325 students	China (Wuhan)	PCL-5, Alcohol use, Tobacco use, PHQ-9, GAD-7, ISI	8.5% have PTSS, 5.9% have alcohol consumption, PTSSs increased if alcohol consumption (16.6% if alcohol, vs. 7.9% if no alcohol).

Abbreviations: AUDIT Alcohol Use Disorder Identification Test, AUDIT-C Alcohol Use Disorder Identification Test-Concise, GAD-7 Generalized Anxiety Disorder-7, ISI Insomnia Severity Index, PCL-5 PTSD Checklist for DSM-5, PHQ Patient Health Questionnaire.

## Data Availability

Not applicable.

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
