# Peer review of "The Intertwining of Posttraumatic Stress Symptoms, Alcohol, Tobacco or Nicotine Use, and the COVID-19 Pandemic: A Systematic Review"

_ijerph, 2022, doi:10.3390/ijerph192114546_

Round 1

Reviewer 2 Report

I am typically as fan of meta-analyses as I think they sum up a good deal of research succinctly.  While this study does that, my concern is that this is examining a "niche" topic that may/may not be relevant generally. The relationships between PTSS and drug use are well established, and simply looking at the effect of adding another, although rare, stressor seems not to add much to the literature in general. But that's simply my opinion - the authors did use good methodology (e.g., PRISMA guidelines, a clear definition of exclusion/inclusion criteria etc.).

One note - there is a spelling error in line 31 in the abstract
